# Volatile Organic Compound (VOC) Profiles of Different *Trichoderma* Species and Their Potential Application

**DOI:** 10.3390/jof8100989

**Published:** 2022-09-21

**Authors:** Liberata Gualtieri, Maurilia Maria Monti, Francesca Mele, Assunta Russo, Paolo Alfonso Pedata, Michelina Ruocco

**Affiliations:** 1Institute for Sustainable Plant Protection (CNR-IPSP), Piazzale Enrico Fermi 1, 80055 Portici, Naples, Italy; 2Department of Agricultural Sciences, University of Naples Federico II, 80055 Portici, Naples, Italy

**Keywords:** volatilome, *Trichoderma*, soil–microbe interactions, volatile organic compounds (VOCs), PTR-Qi-TOF-MS

## Abstract

Fungi emit a broad spectrum of volatile organic compounds (VOCs), sometimes producing species-specific volatile profiles. Volatilomes have received over the last decade increasing attention in ecological, environmental and agricultural studies due to their potential to be used in the biocontrol of plant pathogens and pests and as plant growth-promoting factors. In the present study, we characterised and compared the volatilomes from four different *Trichoderma* species: *T. asperellum* B6; *T. atroviride* P1; *T. afroharzianum* T22; and *T. longibrachiatum* MK1. VOCs were collected from each strain grown both on PDA and in soil and analysed using proton transfer reaction quadrupole interface time-of-flight mass spectrometry (PTR-Qi-TOF-MS). Analysis of the detected volatiles highlighted a clear separation of the volatilomes of all the four species grown on PDA whereas the volatilomes of the soil-grown fungi could be only partially separated. Moreover, a limited number of species-specific peaks were found and putatively identified. In particular, each of the four *Trichoderma* species over-emitted somevolatiles involved in resistance induction, promotion of plant seed germination and seedling development and antimicrobial activity, as 2-pentyl-furan, 6PP, acetophenone and *p*-cymene by *T. asperellum* B6, *T. atroviride* P1, *T. afroharzianum* T22 and *T. longibrachiatum* MK1, respectively. Their potential role in interspecific interactions from the perspective of biological control is briefly discussed.

## 1. Introduction

Volatile organic compounds (VOCs) represent a small, but vital portion of the total metabolites produced by living beings derived from both primary and secondary metabolism and characterised by a low molecular weight, low boiling point and high vapor pressure. Their unique properties enable them to mediate important ecological multi-organism interactions both below and above ground, inducing a wide spectrum of responses [1]. VOCs in general, and those emitted by fungi in particular, consist of molecules from different classes. Fungal VOCs belong to several chemical groups with different biochemical origins such as monoterpenes, sesquiterpenes, alcohols, aldehydes, aromatic compounds, esters, furans, hydrocarbons and ketones as well as nitrogen- and sulphur-containing compounds [2,3]. These compounds may play important roles in inter-and intra-individual communication involving plants, antagonists and mutualistic symbionts [4]. In plants, fungal VOCs are involved in the biocontrol of phytopathogens and pests [5,6,7], acting as attractants or deterrents for insects and other invertebrates or activating plant defence responses against a pathogen attack as well as providing growth promotion [8,9,10,11].

To date, a complete picture of the role of fungal VOCs is still lacking; therefore, the characterisation of species-specific volatile profiles would be helpful to unravel their ecological functions [12,13,14,15].

Fungal species belonging to *Trichoderma* spp. are common soil-borne fungi and important opportunistic avirulent plant symbionts and parasites of other fungi as well as beneficial microorganisms in the agro-ecosystem; they are able to influence the soil health and crop performance [16]. The genus *Trichoderma* includes 254 identified species [17] ubiquitously present in forest and agricultural soils, where they are highly interactive with plant roots and rhizospheric microorganisms [18]. Thanks to this wide range of effects, *Trichoderma* spp. are largely used as biocontrol agents (BCAs) and plant growth promoters. In particular, the VOCs emitted by *Trichoderma* spp. have a strong effect against plant pathogenic fungi such as *Sclerotinia sclerotiorum*, *Sclerotium rolfsii*, *Fusarium oxysporum, Ganoderma* sp., *Penicillium oxalicum, Stagonosporopsis cucurbitacearum*, *Alternaria panax, Botrytis cinerea, Cylindrocarpon destructans* and *Sclerotinia nivalis* [19,20,21].

The volatile profile emitted by *Trichoderma* species can considerably change [12,13,14], not only depending on the species, but also as a consequence of the interaction with other organisms [15,22], suggesting important ecological functions of VOCs [8,23,24]. Among the VOCs reported to be emitted by *Trichoderma* [14], hydrocarbons, heterocycles, aldehydes, ketones, alcohols, phenols, thioalcohols and thioesters and their derivatives [25] are the most represented. C8 compounds such as 1-octen-3-ol and 3-octanone (both of them responsible for the mushroom flavour) that are end-products of fatty acid metabolism, have been shown to play a role in the biocontrol activity of *Trichoderma* spp. [8,23,24] and are already used in biological control practices as fungistatic and fungicidal molecules [26,27]. 6-pentyl-alpha-pyrone (6PP) (a lactone with a coconut-like aroma), which is produced by different *Trichoderma* species such as *T. atroviride* [28,29], *T. asperellum* [30], *T. viride* [31], *T. harzianum* [32], *T. koningii* [33], *T. citrinoviride* and *T. hamatum* [34], has been reported to be able to increase root branching and root hair development [30,35] and to have an effect on plant growth and health at different concentrations [13].

GC-MS is one of the most utilised techniques for VOC detection, even though it has limitations such as its cost-effective separation, identification and quantification of substances combined with its destructive protocols and gas concentration steps [8,12,36]. Despite these limitations, the GC-MS technique has been utilised for VOC profile analyses such as for the “white truffle” fungus *Tuber magnatum* [37] and in studies on VOC-mediated interspecific interactions in the soil, in particular from different species belonging to the *Fusarium* genus [38] and bacteria [39].

Proton transfer reaction time-of-flight mass spectrometry (PTR- -TOF-MS) has recently been introduced to measure volatile emissions from plants [40,41], soils [42,43], yeasts [44] and bacteria and fungi [14,37,45,46,47,48,49], allowing researchers to overcome a few of the GC-MS limitations. This modern technique for the real-time monitoring of VOCs is highly sensitive and can detect low concentrations of VOCs (parts per trillion volume (pptv)) in air and gas samples, providing a rapid, non-invasive fingerprinting of VOC profiles [50].

In the present work, we characterised the volatile profiles of four different *Trichoderma* species—*T. longibrachiatum* MK1, *T. atroviride* P1, *T. asperellum* B6 and *T. afroharzianum* T22—grown both on PDA and in soil by a PTR-Qi-TOF-MS analysis in order to define the VOC molecular markers for each species useful to detect their presence. Microbial volatile detection and characterisation may be used as a diagnostic tool. Moreover, characterising the VOC profile emitted by each different *Trichoderma* species and their role in plant interactions could greatly increase the potential of *Trichoderma* use in agriculture, becoming a tool for the screening and identification of other beneficial microorganisms.

## 2. Materials and Methods

### 2.1. Fungal and Soil Sampling

Four different *Trichoderma* species—*T. asperellum* B6 [51], *T. atroviride* P1 [52], *T. afroharzianum* T22 [53] and *T. longibrachiatum* MK1 [54]—were analysed (Figure 1a). Each *Trichoderma* species was grown on potato dextrose agar (PDA) in 100 mm × 15 mm Petri dishes and maintained at 25 ± 1 °C in the dark with >80% humidity starting from stocks in 20% glycerol stored at −80 °C. All isolates were part of the CNR-IPSP collection. For the VOC measurements, the fungal species were inoculated both on PDA and in soil. The PDA inoculation was performed with 0.5 cm mycelium fungal plugs and transferred into a 1 L Erlenmeyer conical glass flask that contained 100 mL of PDA and equipped with a GL45 3-valve screw cap (Figure 1b). The soil inoculation was carried out with 5 mL of a spore suspension (1 × 10^8^ spores/mL) of each fungal species in 100 mL of non-sterile commercial soil (Universal potting soil-Floragard Vertriebs-GmbH Oldenburg) contained in a 1 L Erlenmeyer conical glass flask equipped with a GL45 3-valve screw cap. The VOCs from PDA and non-sterile commercial soil contained in the same type of flasks without an inoculum were used as control. All samples were incubated at 25 °C in darkness conditions.

### 2.2. VOC Analyses

#### 2.2.1. Mass Spectrometer Analysis of the VOCs Produced by *Trichoderma* spp. Growing on PDA and in Soil

The VOCs emitted by the *Trichoderma* species were measured using PTR-Qi-TOF-MS equipment (Ionicon Analytik GmbH, Innsbruck, Austria) in an air-conditioned room with a constant temperature of 25 ± 1 °C. The protonation of VOCs was carried out using H_3_O^+^ as a proton donor in the transfer reaction and was effective for VOCs with a proton affinity higher than that of H_2_O (691.7 kJ mol^−1^). The headspace VOC profiles accumulated in the flasks described in Section 2.1 were measured by a direct injection of the volatile mixture into the PTR-Qi-TOF-MS drift tube via a heated (80 °C) PEEK inlet tube connected to a valve of the GL45 3-valve screw cap. A flow rate of 100 sccm (standard cubic centimetres per minute) in a range of 20–300 *m/z* for 600 s with an acquisition rate of one spectrum per second was used. The drift tube conditions were 3.8 mbar of pressure, 80 °C temperature and 1000 V drift voltage, resulting in a field density ratio of E/N (with E corresponding with the electric field strength and N with the gas number density) of 141 Td (Townsend: 1 Td = 10^−17^ V cm^2^). A total of three different biological replicates for each sample were analysed; the measurements were taken eight days post-inoculum, when all four fungal species grown on PDA were at the beginning of exponential hyphal growth, a stage during which most secondary metabolites of fungi are produced [55]. The same timing was applied to the soil samples.

#### 2.2.2. PTR-Qi-TOF-MS Data Analyses

The raw data were acquired by TOFDAQ Viewer^®^ software (Tofwerk AG, Thun, Switzerland) and the mass spectra and temporal ion signal profiles were extracted using PTR-MS Viewer software (Ionicon Analytik version 3.3.8) with a custom modified Gaussian function fit for each peak. The data acquisition and peak quantification were expressed as normalised parts per billion by volume (ppbv). To guarantee a high mass accuracy, the calibration of the PTR spectra was performed offline at three calibration points: *m/z* = 21.022 (H_3_O^+^); *m/z* = 203.943 (a fragment of the internal gas standard 1,3-diiodobenzene); and *m/z* = 330.848 ( the internal gas standard 1,3-diiodobenzene). The peaks associated with the PTR-MS ion source—including those ascribed to water chemistry or other interfering ions, e.g., *m/z* = 31.022 (NO^+^), *m/z* = 32.990 (O_2_^+^), *m/z* = 21.022, *m/z* = 37.028 and *m/z* = 39.033 (corresponding with H_3_^18^O^+^ and water cluster ions H_2_O-H_3_O^+^ and H_2_O-H_3_^18^O^+^, respectively)—were eliminated. The *m/z* signals were background-corrected by subtracting the signal obtained from the glass flasks containing only the PDA or the commercial soil. Most of the mass peaks were tentatively identified based on the available literature or by comparisons with genuine standards.

### 2.3. Statistical Analyses

All statistical analyses were carried out by using the Metaboanalyst platform (https://www.metaboanalyst.ca accessed on 20 July 2022) [56]. The data were normalised and autoscaled (mean-centred and divided by the standard deviation of each variable) prior to each analysis. Principal component analysis (PCA) was carried out on both the whole dataset and the two datasets separately (PDA and soil) as an unsupervised method to highlight the underlying data structure. One-way analysis of variance (ANOVA) was performed coupled with Tukey’s HSD test to discover the significantly different means in the multiple comparisons. The ANOVA results were presented by heat maps and hierarchical clustering in order to provide a more intuitive visualisation of the VOC patterns. The rows and columns were reordered so that rows (and columns) with similar profiles were closer to one another, with each entry displayed as a colour related to its signal intensity. Moreover, dendrograms were created using Pearson correlation-based distances and Ward’s method of agglomeration.

## 3. Results

### VOC Analyses

A total of 69 VOCs were detected in the range of the measured masses (mass protonated range *m/z* = 20–300) after the subtraction of the peaks associated with the PTR-MS ion source and their isotopes (Appendix A). The putative identification of the VOCs is reported in Appendix A.

When comparing the total amount of VOCs emitted by the four species on PDA, it was found that, in our conditions, *T. asperellum* B6 emitted the lowest quantity (4.06 × 10^2^ ppbv), followed by *T. afroharzianum* T22, *T. atroviride* P1 (both 4.46 × 10^2^ ppbv) and *T. longibrachiatum* MK1 (4.61 × 10^2^ ppbv). The total VOC emission of *T. asperellum* B6 was significantly lower than all the other species (*p* < 0.05).

In the soil samples, the total VOC emission profiles were very similar among all *Trichoderma* species, with 3.76 × 10^2^ ppbv emitted from *T. asperellum* B6, 3.80 × 10^2^ ppbv from *T. longibrachiatum* MK1, 3.81 × 10^2^ ppbv from *T. afroharzianum* T22 and 3.88 × 10^2^ ppbv from *T. atroviride* P1; the differences were statistically not significant (*p* < 0.05).

A preliminary statistical analysis of the samples from both datasets (PDA and soil) was carried out via principal component analysis (PCA) in order to detect patterns in the measured data without any a priori assumption of a particular distribution of the data. This unsupervised method clearly highlighted the separation of VOCs emitted from *Trichoderma* spp. grown in soil and those grown on PDA (Figure 2), with the three first principal components accounting for 87.0% of the variation in the dataset. It was evident how soil samples formed a more compact group compared with the PDA samples (all data PCA loadings are reported in Appendix A).

Among the PDA samples, *T. longibrachiatum* MK1 and *T. asperellum* B6 were clearly separated by the first two PCs whereas *T. afroharzianum* T22 and *T. atroviride* P1, even if partially overlapping with PC1 and PC2, could be completely distinguished with PC1 and PC3. All soil samples overlapped and the *Trichoderma* species could not be distinguished based on the total VOC emission.

The VOC emission of the four *Trichoderma* species grown on PDA or in soil were then analysed as a separate dataset in order to have a deeper comprehension of the VOC patterns. The PCA of the *Trichoderma* species grown on PDA clearly highlighted the separation into four different clusters, confirming the existence of a distinct volatile blend profile for each species (Figure 3). The first two principal components explained 80.4% of the total variance; with the third component, the total variance explained was 94.7%. In the score plot of the first two principal components (Figure 3a), *T. asperellum* B6 and *T. longibrachiatum* MK1 were plotted in separate quadrants; *T. afroharzianum* T22 and *T. atroviride* P1 shared the same one. The separation between *T. afroharzianum* T22 and *T. atroviride* P1 occurred with the third component (Figure 3b). The PDA PCA loadings are reported in Appendix A.

From the PCA of the *Trichoderma* species grown in soil, a separation of *T. atroviride* P1 and *T. asperellum* B6 was evident. *T. afroharzianum* T22 and *T. longibrachiatum* MK1 were superposed both in the PC1–PC2 and PC1–PC3 plot, with the first three principal components explaining 82.6% of the total variance (Figure 4). The soil PCA loadings are reported in Appendix A.

One-way ANOVA and post hoc test identified 66 out of 68 significant peaks (*p* < 0.05) in the PDA data (Appendix A) and 25 out of 68 significant peaks (*p* < 0.05) in the soil data (Appendix A). The hierarchical heat map clusters of the ANOVA-significant volatiles are presented in Figure 5.

The PDA heat map showed well-defined clusters of volatiles characteristically over- or under-emitted among the four different fungal species. In particular, among the over-emitted clusters, *T. asperellum* B6 over-emitted a specific group of volatiles, including *m/z*: 33.03, 139.150, 267.850, 99.071, 157.176, 145.139 and 219.184 (cluster 1 in Figure 5a). *T. longibrachiatum* MK1 over-emitted a specific group of volatiles, including *m/z*: 117.105, 131.124, 171.194, 61.028, 79.075, 90.075 and 247.233 (cluster 2 in Figure 5a) as well as *m/z* 89.05. *T. atroviride* P1 over-emitted *m/z* 167.099 and 115.076 (cluster 3 in Figure 5a) as well as *m/z* 45.99 and *m/z* 59.049. *T. afroharzianum* T22 over-emitted a group of volatiles, including *m/z* 87.080, 223.082, 269.850 and 70.072 (cluster 4 in Figure 5a) as well as *m/z* 127.128. Among the under-emitted VOCs, different clusters could be identified in *T. asperellum* B6 with *m/z* 57.069, 43.054, 71.085, 101.159 and 137.134 (cluster 5 in Figure 5a) and *T. longibrachiatum* MK1 with *m/z* 121.100 and 29.014 (cluster 6 in Figure 5a); in *T. atroviride* P1, we could only distinguish *m/z* 33.993 and in *T. afroharzianum* T22, there were no characteristic under-emitted VOC clusters.

The VOCs detected from the soil samples treated with the four different *Trichoderma* species allowed a good discrimination between *T. atroviride* P1 and *T. asperellum* B6 whereas *T. afroharzianum* T22 and *T. longibrachiatum* MK1 showed very similar VOC blends. In particular, two VOC clusters were over-emitted in *T. atroviride* P1: *m/z* 99.071, 45.033, 83.083, 33.033 and 33.093 (cluster 1 in Figure 5b) and *m/z* 117.106, 81.070, 115.076, 47.051 and 247.233 (cluster 2 in Figure 5b). Among the under-emitted VOC clusters, we could distinguish *m/z* 107.085 and 75.042 in *T. atroviride* P1 (cluster 3 in Figure 5b,) and *m/z* 59.048, 87.080, 127.128, 43.018 and 70.072 in *T. asperellum* B6 (cluster 4 in Figure 5b).

Table 1 and Table 2 report the top twenty most abundant compounds (Top 20) detected in the VOC blends emitted by the four different species when grown in PDA and soil, respectively.

A high number of the 20 most abundant VOCs emitted was shared among *T. longibrachiatum* MK1, *T. atroviride* P1 and *T. afroharzianum* T22, with few exceptions. In the Top 20 *T. longibrachiatum* MK1 VOCs, three compounds (*m/z* 135.117, 90.075 and 81.070) were specific; in the Top 20 *T. atroviride* P1 VOCs, only *m/z* 167.099 was specific; and in the Top 20 *T. afroharzianum* T22 VOCs, two compounds (*m/z* 95.046 and 121.064) were specific. A different behaviour was shown by *T. asperellum* B6, which had 7 specific compounds out of 20 (*m/z* 33.033, 145.139, 139.150, 219.184, 111.046, 69.069 and 157.176).

There were no significant differences between the Top 20 VOCs emitted by the four different species in the soil samples. The five most abundant compounds were the same for all soil samples analysed: *m/z* 29.014, 33.033, 39.033, 33.993 and 28.050. Among them, the mass *m/z* 29.014 had the highest emission percentage for all VOC blends analysed.

## 4. Discussion

In the present study, the PTR-Qi-TOF-MS technique was utilised to discriminate different species of *Trichoderma*. Our case study was represented by *T. longibrachiatum* MK1, *T. atroviride* P1, *T. asperellum* B6 and *T. afroharzianum* T22 grown on PDA or in soil, looking for a valid strategy of high-throughput screening for *Trichoderma* spp. identification.

This technique was chosen because it is highly sensitive and can detect in real-time low concentrations of VOCs in air and gas samples without sample preparation, derivatisation or concentration; it is excellent for the detection of low molecular weight, oxygenated and polar compounds. The major downside of PTR-Qi-TOF-MS is the identification of compounds, as each detected mass can either be associated with parent molecules or possible fragments from other molecules. Thus, the identification of compounds measured by PTR-Qi-TOF-MS is either putative and based on literature references or determined by a comparison with standards. This mass spectrometry technique has successfully been used to characterise the VOCs of fungi such as *Fusarium* spp. [46], *Muscodor albus* [57], *Tuber magnatum* [37] and different *Mortierella* species [49].

VOCs are involved in various biological processes, including communication among organisms such as plants [22] and microbes as well as in self-signalling [58]. The potential of fungal volatiles is receiving growing attention in agricultural, environmental and ecological studies. Fungal VOCs induce both positive and negative effects on plant growth [13] and have often been used to suppress pathogenic bacteria and fungi [19,59]. Recent studies have reported that microbial VOC compositions are variable, depending on intra- and interspecific interactions [39,60,61]. Previous studies on *Trichoderma* spp. VOCs have shown that the emission profile is species- and even strain-specific as well as substrate composition- and cultivation environment-dependent [15,29,62]. The soil effect on fungal VOC emissions has been previously investigated utilising the solid phase microextraction (SPME) technique coupled with the GC-MS technique [38]; the authors reported deep differences in the VOC profiles of the same fungal species grown in soil or a malt extract medium. In the present study, we characterised *Trichoderma* VOC emissions both in soil and on PDA in order to develop a fast method that reproduced natural field conditions and that could be used as a diagnostic tool for the identification of *Trichoderma* spp. in vivo.

When analysing the PTR-Qi-TOF-MS dataset, we were able to detect 68 VOCs, 66 and 25 of which were differentially produced by the four *Trichoderma* species grown on PDA or in soil, respectively. It is interesting to note that in our experimental condition, *T. asperellum* B6 was significantly the weakest VOC emitter in terms of total ppbv when grown on PDA, whereas the other three species emitted a very similar total amount of VOCs. To the best of our knowledge, this is the first time that such a difference in the total VOC emission has been reported. Even if among *T. longibrachiatum* MK1, *T. atroviride* P1 and *T. afroharzianum* T22 there were no differences in terms of the total VOC emission, their blends of volatiles were diverse in terms of composition. The PCA of the PDA samples (Figure 3) showed a clear separation of four different clades that corresponded with the tested fungal species, allowing us to conclude that different types and quantities of VOCs are produced depending on the fungal species when grown on PDA. The PCA carried out on the whole dataset (both data from PDA and from soil) (Figure 2) clearly highlighted the separation of the volatile profiles of the soil samples from the PDA samples, indicating a significant role of the fungal growth environment on VOC emissions. However, it should be noted that although the profiles of the VOCs obtained from *Trichoderma* spp. grown on PDA were always clearly distinct from each other, those obtained from the fungi grown in soil (Figure 4) permitted us to discriminate well only *T. atroviride* P1 and *T. asperellum* B6 in contrast *T. afroharzianum* T22 and *T. longibrachiatum* MK1 were superposed. In general, all soil samples emitted a minor total VOC amount compared with the PDA samples, probably due to VOC entrapment in the soil or because a large fraction of the compounds produced by the *Trichoderma* species were metabolised by the autochthonous microbiota of the soil. This result was in accordance with Asensio et al. [63] who proved, in a study on the behaviour of Mediterranean soil, that there was an overall VOC uptake. It is known that the soil microbiota and VOC retention in soils are influenced by many environmental factors such as pH, temperature and moisture content [64]. The pH of soils determines the charge of VOCs and modifies their evaporation pressure, which also depends on temperature [65]. Most microbial VOCs are produced in the cell or are released from substrates that are digested by extracellular enzymes, so they are produced in the liquid phase and their emission depends on humidity; in particular, polar compounds are retained more strongly than aromatic and aliphatic molecules [66]. In a few cases, the soil texture can lead to an absorption of VOCs [66] by microorganisms that can utilise them as a carbon source, impacting on the global soil volatilome [67,68]. Moreover, as soil is sugar-poor and rich in amino acids and lipids [69] compared with PDA, which is a sugar-rich medium, the differences in the total volatile amount among the same species grown in soil or on PDA may be related to the different fungal growth rates on the two substrates [70].

The PCA results matched the heat map results where the three replicates of each treatment lay close together, but the four treatment classes were separated. Each species, in particular in the PDA samples, strongly over- or under-emitted characteristic clusters of volatiles, but the relationships among the four VOC blends did not correspond with the phylogenetics of the *Trichoderma* species [71]. Several VOCs produced by the four *Trichoderma* species were shared among them, making the use of a single molecule as a diagnostic tool unfeasible. More useful for this purpose could be the relative abundance of the different volatiles, with the use of the complete profile instead of a selected ion as a fingerprint of a species [72].

From a general analysis of the VOCs emitted by the four *Trichoderma* species, it is interesting to note that many of the identified molecules have been previously reported to play a role during fungi interkingdom communication. Among them, several have been related to resistance induction, promotion of plant seed germination and seedling development and antimicrobial activity, as 2-pentyl-furan [73], 6PP [74], acetone [75], octanol, 1-octen-3-ol and trans-2-octenal [13,23].

A volatilome analysis of each single species was also performed by ranking the Top 20 emitted volatiles in terms of ppbv expressed as percentage of the total emitted VOCs. This approach confirmed the results obtained with the PCA and heat map analyses, detecting *T. asperellum* B6 as the most different in terms of VOC emissions compared with *T. longibrachiatum* MK1, *T. atroviride* P1 and *T. afroharzianum* T22, which had a more similar Top 20 VOC composition.

From Table 1, it was evident that the VOC profiles of the four different *Trichoderma* spp. differed more in quantitative than in qualitative terms. The species-specific mass peaks could not be defined as being exclusively produced by a species, rather, as those with significantly higher concentrations. In the Top 20 emitted compounds, we noticed a few masses shared among all PDA samples such as an alkyl fragment (*m/z* 41.039), acetic acid fragment (*m/z* 43.017) and ethanol (*m/z* 47.050). *T. asperellum* B6 in our experimental condition (PDA) emitted seven compounds not present in the Top 20 of the other three species; in particular, *m/z* 33.033 (corresponding with methanol) was the most abundant (2.02%). Methanol production has been proven for several *Fusarium* species grown on PDA [46]; however, up to now, it has never been reported for *Trichoderma* species. It is interesting to note that in the soil experiments, methanol was present as one of the Top 20 VOCs for all four species, probably being produced by the fungi growth on the plant decay material present in the soil samples [43,76]. Another interesting molecule among the Top 20 emitted VOCs of *T. asperellum* B6 was the compound with *m/z* 139.150, putatively identified as 2-pentylfuran (Appendix A). 2-pentylfuran has been correlated with a reduction in downy mildew severity on grapevines when applied without physical contact with the leaf tissues [73]. Among the most emitted Top 20 VOCs, *p*-cymene (*m/z* 135.117) is *T. longibrachiatum* MK1-specific. Similar results were obtained by Guo et al. [14], who compared *T. hamatum* with three other *Trichoderma* species. *p*-cymene is a monoterpene known for its antibacterial, antiviral and antifungal activities [77]. Characteristic of the Top 20 *T. atroviride* P1 VOCs was 6-pentyl-alpha-pyrone (6PP) (*m/z* 167.099). This volatile is responsible for the coconut-like scent of *T. atroviride* isolates growing on PDA [78] and has been correlated with antimicrobial activity and plant defence induction as well as plant growth promotion [74,79,80]. Among the *T. afroharzianum* T22 Top 20 emitted VOCs, there was *m/z* 121.064, putatively identified as acetophenone, a molecule reported to show antifungal activity in vitro against *Penicillium italicum* [81]. The four *Trichoderma* species grown on PDA, even if sharing a few compounds among the Top 20 emitted VOCs, showed characteristic molecules that could all be reconducted to the broad-spectrum activities and multiple modes of action of *Trichoderma* spp.

In all Top 20 soil sample-emitted VOCs, the four volatile profiles showed a similar amount of emitted compounds; the most abundant VOCs were mainly represented by low *m/z* VOCs, led by methanol (*m/z* 33.033). This phenomenon could be related to the metabolic activities of the soil microbial biomass as respiration and enzyme activities which are responsible for the generation of short-chain VOCs and fragments.

## 5. Conclusions

A complete knowledge of the volatile profiles of the different *Trichoderma* species and the relative quantities of each VOC are fundamental prerequisites for the characterisation of the different species, for their detection in vivo as well as real-time studies on their multi-trophic interactions. Our study revealed significant quantitative volatilome differences among the four analysed *Trichoderma* spp. grown on PDA, allowing the characterisation of the different volatile blends. The PTR-Qi-TOF-MS technique was found to be suitable for quickly surveying in vivo the different fungal species within the samples growing on PDA, but it failed to discriminate among them when grown in soil and to identify volatile molecular markers. The potential of using PTR-Qi-TOF-MS for the in vivo characterisation of soil microbial communities could be fruitfully exploited, integrating data from other different sources to train artificial intelligence-based systems, resulting in the definition of precise and accurate models.

## Figures and Tables

**Figure 1 jof-08-00989-f001:**
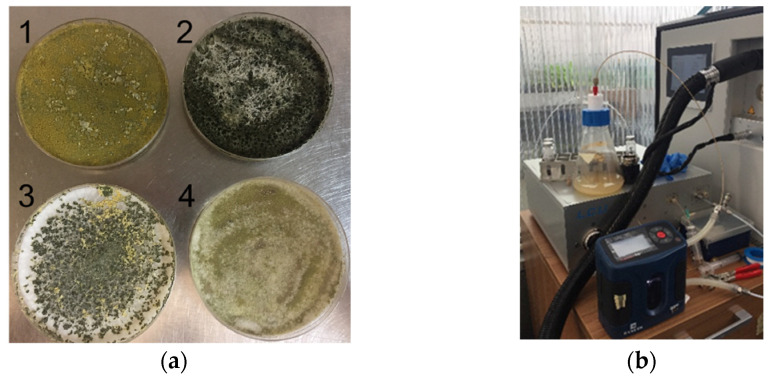
(**a**) Petri dishes containing *Trichoderma* isolates grown at 25 ± 1 °C in the dark for 14 days: 1: *T. longibrachiatum* MK1; 2: *T. asperellum* B6; 3: *T. afroharzianum* T22; 4: *T. atroviride* P1; (**b**) Erlenmeyer conicalglass flask equipped with a GL45 3-valve screw cap containing 100 mL of potato dextrose agar (PDA) medium, connected online with PTR-Qi-TOF-MS.

**Figure 2 jof-08-00989-f002:**
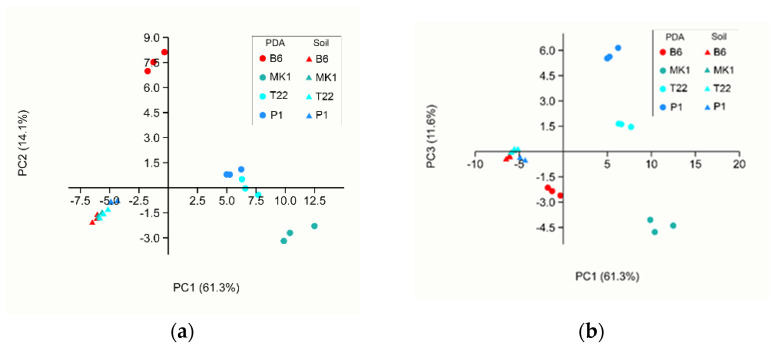
First three components of the PCA analysis of the PTR− Qi− TOF− MS data of VOCs emitted in the samples from PDA and soil: (**a**) PC1 and PC2 plot; (**b**) PC1 and PC3 plot. The variance explained by each component is reported in brackets.

**Figure 3 jof-08-00989-f003:**
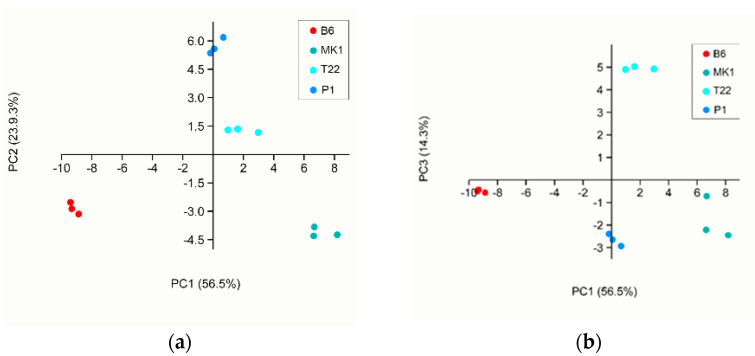
First three components of the PCA analysis of the PTR-Qi-TOF-MS data of VOCs emitted by samples grown on PDA: (**a**) PC1 and PC2 plot; (**b**) PC1 and PC3 plot. The variance explained by each component is reported in brackets.

**Figure 4 jof-08-00989-f004:**
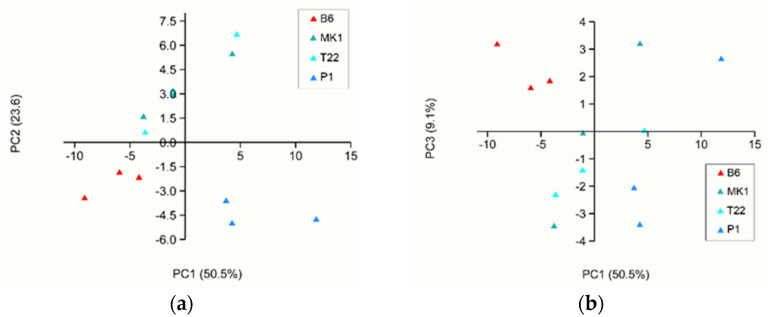
First three components of the PCA analysis of the PTR-Qi-TOF-MS data of VOCs emitted by samples grown in soil: (**a**) PC1 and PC2 plot; (**b**) PC1 and PC3 plot. The variance explained by each component is reported in brackets.

**Figure 5 jof-08-00989-f005:**
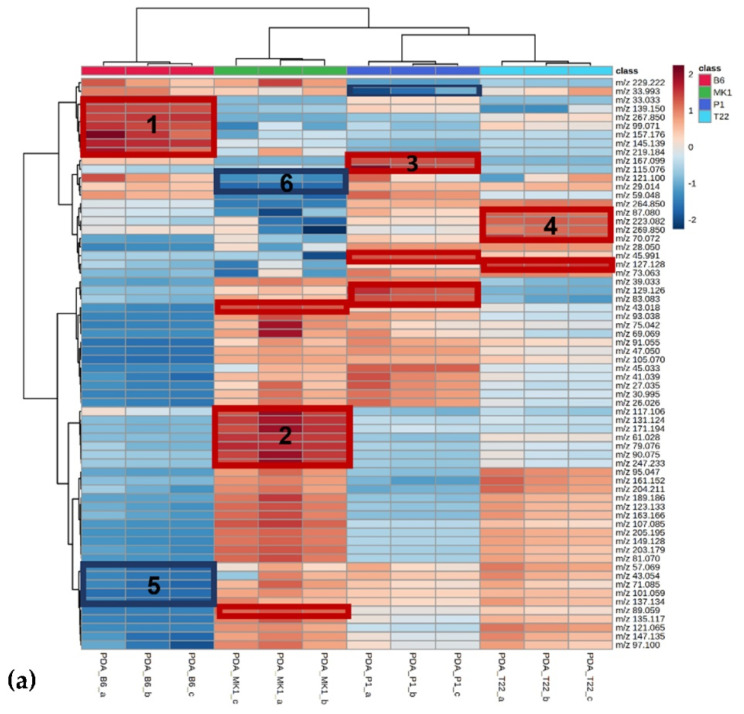
Heat maps and two-dimensional hierarchical dendrograms of VOCs emitted during fungal growth on PDA (**a**) and soil (**b**). Replicates are in columns and variables are in rows. Each coloured cell on the map corresponds with a concentration value following a blue/red chromatic scale from −2 value (very low expression) to 2 (extremely high expression). Numbered boxes represent well-defined VOC clusters characteristically over (red)- or under (blue)- emitted. Only ANOVA-significant peaks are used (*p* < 0.05). The Pearson distance and Ward’s clustering algorithm were used for the dendrograms. The probable identification and chemical group of the VOCs are detailed in Appendix A.

**Table 1 jof-08-00989-t001:** Emissions of the Top 20 VOCs of *Trichoderma* isolates grown on PDA.

B6 ^1^	%VOCs ^2^	MK1 ^1^	%VOCs ^2^	P1 ^1^	%VOCs ^2^	T22 ^1^	%VOCs ^2^
29.014	2.23	43.018	2.13	47.051	2.21	29.014	2.04
59.048	2.07	47.051	2.10	45.033	2.15	47.051	2.01
33.033	2.02	61.028	2.06	29.014	2.05	205.195	1.93
41.039	1.87	89.059	2.01	41.039	1.98	57.069	1.91
43.018	1.82	205.195	2.00	43.018	1.97	43.018	1.90
33.993	1.81	93.038	1.88	59.048	1.95	41.039	1.85
57.069	1.79	41.039	1.86	93.038	1.85	43.054	1.77
145.139	1.78	45.033	1.83	57.069	1.85	61.028	1.75
47.051	1.77	29.014	1.81	131.124	1.77	45.033	1.75
139.150	1.73	57.069	1.79	26.026	1.76	89.059	1.72
27.035	1.68	149.128	1.75	27.035	1.75	71.085	1.71
219.184	1.67	71.085	1.73	167.099	1.74	59.048	1.70
61.028	1.66	131.124	1.71	105.070	1.73	73.063	1.69
111.046	1.64	135.117	1.69	45.991	1.73	95.047	1.68
26.026	1.64	105.070	1.69	43.054	1.73	149.128	1.68
73.063	1.63	90.075	1.68	39.033	1.70	121.065	1.67
69.069	1.62	26.026	1.67	71.085	1.68	33.993	1.64
39.033	1.60	27.035	1.66	73.063	1.67	27.035	1.63
157.176	1.58	39.033	1.65	89.059	1.62	105.070	1.63
45.991	1.57	81.070	1.64	33.993	1.61	26.026	1.63

^1^ Protonated measured (*m/z*) VOCs of different *Trichoderma* species. ^2^ The emission percentage (%) calculated as VOCs emitted/total VOCs.

**Table 2 jof-08-00989-t002:** Emissions of the Top 20 VOCs of *Trichoderma* species grown in soil.

B6 ^1^	%VOCs ^2^	MK1 ^1^	%VOCs ^2^	P1 ^1^	%VOCs ^2^	T22 ^1^	%VOCs ^2^
29.014	2.41	29.014	2.39	29.014	2.33	27.035	2.24
33.033	1.99	33.033	1.97	33.033	1.99	28.050	1.97
39.033	1.93	39.033	1.92	33.993	1.91	26.026	1.95
33.993	1.89	33.993	1.88	39.033	1.85	29.014	1.91
28.050	1.87	28.050	1.85	28.050	1.81	30.995	1.87
27.034	1.82	43.018	1.82	99.071	1.81	33.033	1.80
43.017	1.78	27.035	1.79	43.018	1.80	39.033	1.79
26.026	1.76	26.026	1.75	43.054	1.80	41.039	1.75
61.034	1.72	43.054	1.74	41.039	1.77	43.018	1.74
59.048	1.70	61.028	1.73	27.035	1.76	43.054	1.73
41.039	1.69	41.039	1.73	45.033	1.75	45.033	1.72
264.850	1.68	59.048	1.72	26.026	1.71	45.991	1.68
45.991	1.68	264.850	1.67	61.028	1.70	47.051	1.67
43.054	1.67	45.991	1.64	45.991	1.70	57.069	1.64
47.050	1.66	47.051	1.63	59.048	1.70	59.048	1.60
30.995	1.63	30.995	1.63	264.850	1.67	61.028	1.58
204.211	1.59	73.063	1.60	47.051	1.65	69.069	1.58
45.033	1.57	45.033	1.58	30.995	1.59	70.071	1.54
73.063	1.57	204.211	1.58	81.070	1.58	73.063	1.52
69.068	1.54	69.068	1.54	204.211	1.54	81.070	1.46

^1^ Protonated measured (*m/z*) VOCs of different *Trichoderma* species. ^2^ The emission percentage (%) calculated as VOCs emitted/total VOCs.

## Data Availability

Not applicable.

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
