# Peer review of "Volatile Organic Compound (VOC) Profiles of Different Trichoderma Species and Their Potential Application"

_jof, 2022, doi:10.3390/jof8100989_

Round 1

Reviewer 1 Report

The present manuscript analyzed and evaluated the VOCs profiles of different Trichoderma species and their potential application. The research is original, and some important details and information are provided in the present manuscript, but the current version need to supply much more work on the species of Trichoderma identification and the function of the emission of Top VOCs analysis in different isolates.

Please see my comments below for specific questions/suggestions I had.

Specific comments:

Lines 19-23: The statements of abstract made are very ambiguous and do not provide a summary of the main results obtained. Also, please combine or delete some of the methos description in the abstract.

Lines 10-11: Please provide the specific information of four strains of Trichoderma. Did you identify the species of Trichoderma strains before this experiment conduction (Four different Trichoderma species T. asperellum B6, T. atroviride P1 [48], T. afroharzianum T22 [49] and T. longibrachiatum MK1 [50], were analyzed)?

Lines 120-125: Two or more sequence sites should be applied to identify the Trichioderma species, and also it is difficult to identify the specific species using the ITS region. Normally, the specific species of Trichioderma identification should be based on the molecular identification and morphological characterization. Why identify the species of Trichoderma strains if you have identified the species in lines 10-11?

Line 137: Please change “Trichoderma” to “Trichoderma”.

Much more work should mainly focus on analyze the function of the emission of Top VOCs of Trichoderma isolates.

Reviewer 2 Report

There are some minor points need to be addressed:

1. Some introduction parts need citation (Line 45-52)

2. Materials and methods 2.1 commercial soil ? is there sterile soil? if not maybe VOCs from other microorganism may interupt this analysis.

3. Materials and methods 2.2 All Trichoderma species has been identified into species level why in this study you need to identify again?

Some minor points can be found in the pdf. file

Reviewer 3 Report

Authors analyse the VOCs emitted by four Trichoderma species after growing on PDA or soil. Differences were detected in the quantity of VOCs from the four species grown on PDA while VOCs from soil-grown fungi were similar.

The manuscript provides interesting information and the results are worth being published. However, I would like to give some specific comments of points to be improved:

-    -Methods: Authors confirm the identity of the four Trichoderma species used in this work. I think it is not necessary but, in that case, authors should do it by sequencing of two different markers, not only an ITS region.

-     -Quality of figures 3, 4, 5 and 6 needs to be improved (especially the legends).

-     -Reference section should be reviewed. There are lots of mistakes.

-     -Line 100: remove “The” (The 2.1 Fungal and soil sampling)

-   -Line 116: Indicate the growth temperature and growth time of Trichoderma strains in figure 1 (a).

Reviewer 4 Report

Please note some points raised in the attached comments file. This technique sounds very good to study other fungi.

Comments on jof 1860140

·        Title, Volatile Organic Compounds (VOCs)…

·        Lines 45-48 could be deleted. Also, 56-58.

·        Note, contents of lines 60-62 not discussed as far as which compounds discovered

·        65-75 kinds of names of those compounds not contrasted compered to findings.

·        “last decades” change

·        Lines 101-102, were those isolates obtained from those references.

·        Line 107, rephrase. 0.5cm plugs…

·        Did you mean “Erlenmeyer Conical flask”.

·        There should be a control flask with PDA only.

·        “commercial soil” not scientific. What was the components? Was it “live soil” or sterilized? If it was live soil what was the origin? What microbial constituents in that live soil or commercial soil???

·        Fig 1, not clear pictures of the fungal growth on plates. Needs to be coupled with micrographs of those isolates.

·        Molecular identification of those isolates need to be introduced in the material and method section. Lines 186-187

·        It is usually accepted to have the data in the text following the place where is mentioned. Tables mentioned but Figures preceded them. Lines 195-197

·        Fig 2, could be replaced with simple text.

·        Fig 3, 4 and 5 not clearly discussed for what significance those VOC only dealt with?

·        Dendrograms has no values and not discussed.

·        Tables 1, and 2, kind and nature of those VOCs?

·        Conclusions, lines 460-463 very broad and not yet supported assumption.
